

# Measurement of bovine (*Bos taurus*) serum albumin by different methods and the comparison of protein fractions determined by capillary zone electrophoresis and agarose gel electrophoresis

Leandro Abreu da Fonseca[1,2], Fabiano Montiani-Ferreira[2,3], Marilyn Rodriguez[2], Joao Henrique Jabur Bittar[4], Fabricia Modolo Girardi[1], Theo Matos Arantes Moraes[1] and Carolyn Cray[2]

[1] Veterinary Department, Universidade Federal de Viçosa, Vicosa, Minas Gerais, Brazil
[2] Department of Pathology and Laboratory Medicine, University of Miami, Miami, FL, United States of America
[3] Veterinary Medicine Department, Universidade Federal do Paraná, Curitiba, Paraná, Brazil
[4] Large Animal Clinical Sciences Department, University of Florida, Gainesville, FL, United States of America

Corresponding author
Leandro Abreu da Fonseca, leandroabreu@ufv.br

## ABSTRACT

Serum albumin measurement is an important parameter routinely evaluated in clinical biochemistry within the livestock industry. It plays a crucial role in assessing the nutritional and health status of animals, as well as in aiding the diagnosis of various pathological conditions as a complementary tool. Several laboratory methods are available for albumin measurement; however, some methods have been documented to overestimate the concentration of serum albumin *versus* the proposed gold standard of serum protein electrophoresis. The primary aim of the study was to analyze the agreement between albumin measurement in bovine serum samples by the capillary zone electrophoresis (CZE), agarose gel electrophoresis (AGE), bromocresol green (BCG), and purple (BCP) methods. In addition, AGE and CZE methods were also compared for quantitation of globulin fractions. Lastly, reference intervals were established using all methods using the American Society for Veterinary Clinical Pathology (ASVCP) guidelines. Serum samples from 55 clinically normal Brangus cattle ($5 \pm 1.5$ years old) were examined by the four methods. For the albumin method comparison, all methods were significantly correlated ($r = 0.55$–$0.91$, $p < 0.0001$) and the bias between methods ranged from 0.01–0.22 g/dL. For AGE and CZE methods, all protein fractions were significantly correlated ($r = 0.85$–$0.91$, $p < 0.0001$) except for the alpha 1 fraction ($r = 0.21$, $p = 0.12$). Five fractions were quantitated using AGE method: albumin, alpha 1, alpha 2, beta, and gamma globulins. For the CZE method, six fractions were resolved with the sub fractionation of beta 1 and beta 2 globulin fractions. The evaluated methods showed good agreement in determining albumin levels in cattle. Although CZE offers higher resolution, it requires careful interpretation and precise definition of fractions. Therefore, methodological choice and consideration of analyzer-specific reference intervals are essential for accurate results.

## INTRODUCTION

Data obtained from clinical biochemistry testing is an important tool for veterinary clinicians. One of the most common measurements is serum albumin as it can reflect nutritional status; liver, gastrointestinal, and kidney diseases; and the presence of an acute phase response (*Kaneko, 2008*). Several laboratory methods are available for albumin measurement including bromocresol green (BCG) and bromocresol purple (BCP) dyes implemented on automated chemistry analyzers and protein electrophoresis (*Kumar & Banerjee, 2017*). These methods can vary in the type of application (*i.e.,* options for point of care *versus* reference laboratory) as well as specificity for albumin. Differences may not only have clinical implications but also necessitate method and species-specific reference intervals.

In clinical biochemistry analyzers, the BCG and BCP dyes form the basis of the primary methods for the determination of serum albumin (*Brackeen, Dover & Long, 1989*; *Tanaka et al., 2023*). The BCG method is used more commonly in veterinary medicine as it is frequently available on automated chemistry analyzers. However, BCG has been documented to overestimate the concentrations of serum albumin as the dye can also react with globulins in many species including bovine, equine, ovine, reptile, and avian species (*Keay & Doxey, 1984*; *Cray, 2021*; *Filípek & Illek, 2021*). The second common method, utilizing BCP dye, has been reported in studies of human serum to have higher specificity for albumin, and previous reports have found BCP methods produce results in better agreement with immunoassays and protein electrophoresis (*Brackeen, Dover & Long, 1989*; *Duly et al., 2003*; *Kumar & Banerjee, 2017*). However, in animal species, BCP methods were reported to result in lower values in canine, rodent, and bovine species (*Evans & Parsons, 1988*; *Filípek & Illek, 2021*). It is important to note that the BCG and BCP methods implemented on chemistry analyzers are calibrated for the measurement of human albumin; reaction times and calibrators may influence the results of samples from non-human species (*Tóthová, Mihajlovičová & Nagy, 2018*). In addition, another important factor is that species specific structural differences in albumin may affect the ability to bind different dyes (*Filípek & Illek, 2021*).

Serum protein electrophoresis provides a different methodology for quantifying albumin based on the fractionation of proteins and is often considered the gold standard of albumin quantitation in laboratory medicine. Agarose gel electrophoresis (AGE), commonly used in veterinary laboratories, utilizes a protein stain and densitometry allowing for quantitation of albumin and 4–6 globulin fractions (*Tóthová, Mudroň & Nagy, 2017*; *Cray, 2021*). However, both the ability of the stain to bind to protein and the process of densitometry may result in less accuracy. More recently, capillary zone electrophoresis (CZE) has been described for use with animal samples with reported increased resolution and accuracy (*Crivellente, Bonato & Cristofori, 2008*; *Cray, 2021*). In this method, protein fractions are quantitated by an ultraviolet detector (*Cray, 2021*).

The primary aim of the study was to examine the agreement between albumin measurement in bovine serum samples by the CZE, AGE, BCP, and BCG methods. In addition, AGE and CZE methods were also compared for quantitation of globulin fractions. Lastly, reference intervals (RI) were established using all methods.

## MATERIALS AND METHODS

### Samples

The bovine serum samples used in this study were sourced from a cattle farm located in South Florida (St. Lucie County, FL, USA). Samples were collected as part of routine health screening at this farm and an aliquot was utilized for this study. The serum was sourced from clinically healthy Brangus cows that ranged from 3 to 10 years old (mean $\pm$ SD, $5 \pm 1.5$ years), and were non-pregnant. Cattle were kept in grazing pastures of Bahia (*Paspalum notatum*) and Tifton (*Cynodon dactylon*) grasses, with supplementation of minerals, molasses, and water *ad libidum* available year around. All of the experimental procedures were approved by Federal University of Viçosa (UFV) Animal Care and Use Committee (Approval number, 56-2024; December 27, 2024). No euthanasia procedures were performed on the animals. After collecting the samples, the animals were released and followed the farm's daily management routine, without any other interventions or procedures.

The animals were manually restrained and blood samples were collected by coccygeal vessel puncture into a tube without anticoagulants (BD-Vacutainer, Becton Dickson Company, Franklin Lakes, NJ, USA). The sample was stored on ice and transported to the laboratory where centrifugation was performed at $2,000\times$ g for 20 min at room temperature. The serum was aliquoted and stored at $-80\,°C$ until further biochemistry analysis and electrophoresis were performed. Only serum samples free of hemolysis were used, and a total of 55 samples met this criterion and were analyzed.

### Bromocresol green and purple methods

Samples were analyzed on a Vitros 5600 chemistry analyzer (Ortho, Rochester, NY, USA). The analyzer was maintained per the manufacturer's instructions. The BCG method utilized dry slide methodology purchased from and calibrated with reagents also from Ortho. The BCP method utilized wet chemistry reagents purchased and calibrated from Sigma (St. Louis, MO, USA). For both methods, a bovine serum albumin calibrator (VITROS Chemistry Products Calibrator, Ortho, Rochester, NY, USA) was used as an analytical standard.

### Agarose gel electrophoresis

The agarose gel electrophoresis was determined by the SPIFE 3000 system (Helena Laboratories, Inc., Beaumont, TX, USA), and the serum samples were prepared as described by *Fonseca et al. (2023)*. The percentage of area under the curve corresponding to each fraction was determined after scanning and then multiplied by the total protein to determine the absolute protein concentration in each fraction. The albumin/globulin (A/G) ratio was calculated by dividing the albumin by the sum of globulin fractions.
Total protein was analyzed using the biuret method and a chemistry analyzer (Ortho Integrated Systems Vitros 5600; Ortho Vitros Diagnostics, Rochester, NY, USA). Normal and abnormal human control samples (Helena Laboratories, Inc., Beaumont, TX, USA) were also included in the analysis.

## Capillary zone electrophoresis

Serum samples were submitted to Sebia Capillarys 2 Flex Piercing system (Sebia, Norcross, GA) according to the manufacturer's protocols for protein electrophoresis. After the relative values were determined for each fraction, the absolute values were obtained by multiplying the percentages for each fraction by the total protein concentration, as determined using the biuret method and a chemistry analyzer (Ortho Integrated Systems Vitros 5600; Ortho Vitros Diagnostics, Rochester, NY, USA). The A/G ratio was calculated by dividing albumin by the sum of globulin fractions, and human control samples were also used. The samples were also prepared and analyzed as described by *Fonseca et al. (2023)*.

## Statistical analysis

The descriptive and inferential statistical analyses, and the reference intervals were all performed using MedCalc® Statistical Software version 20.027 (MedCalc Software Ltd, Ostend, Belgium) and StatView 5.0 (SAS Institute, Cary, NC, USA), with a significance $p$-value <0.05. The D'Agostino-Pearson test was used to check for normality of distribution. The comparison method was performed as previously described (*Jensen & Kjelgaard-Hansen, 2006*) and included Spearman's or Pearson's correlation, Passing-Bablok regression, and Bland Altman plots. Spearman's correlation ($r_s$ value) was considered very strong (>0.70), strong (0.40−0.69), moderate (0.30−0.39), weak (0.20−0.29), or negligible (0.01−0.19) as previously described. The Cusum test for linearity was performed for each comparison. Pearson's correlation ($r_s$ value) was considered very high (0.90−1.00), high (0.70−0.90), moderate (0.50−0.70), low (0.30−0.50), or negligible (<0.29). The reference intervals were calculated following the American Society for Veterinary Clinical Pathology (ASVCP) guidelines (*Friedrichs et al., 2012*). Resulting $p$-values <0.3 by the D'Agostino-Pearson test were judged as having a non-Gaussian distribution. Outliers and suspect data were determined based on Tukey's test (*Friedrichs et al., 2012*). Outliers were calculated using Tukey's method: outliers are values more than 1.5 times the interquartile range from the quartiles—either below Q1−1.5 IQR, or above Q3 + 1.5 IQR.

## RESULTS

### Method comparison

For albumin method comparison, Passing-Bablok regression and Bland-Altman analysis were performed (Table 1). All methods were significantly correlated ($r = 0.55$–$0.91$, $p < 0.0001$), and the bias between methods ranged from 0.01 to 0.22 g/dL. Albumin values obtained *via* AGE method exhibited a slight positive bias when compared to the BCG method, and a constant error was identified *via* Passing-Bablok regression. Additionally, both BCG and BCP methods slightly overestimated albumin concentrations relative to the CZE method. These differences are illustrated in the Bland-Altman plots (Fig. 1) and Passing-Bablok regression analyses (Fig. 2).

**Table 1** Passing-Bablok regression analysis, Bland Altman analysis, and Spearman's correlation of different measurement methods for serum albumin from cattle; sample size: $n = 55$.

| Albumin method | Spearman's correlation | Passing-Bablok regression | | | | Bland Altman bias |
|---|---|---|---|---|---|---|
| | $r_s$ value (p value) | y-intercept | 95% CI | Slope | 95% CI | Mean (SD) |
| AGE vs. CZE | 0.911 (<0.0001) | 0.22 | −0.02 to 0.43 | 0.97 | 0.90–1.06 | 0.17 (0.12) |
| AGE vs. BCG | 0.788 (<0.0001) | 0.53 | 0.15 to 0.77[a] | 0.89 | 0.78–1.01 | 0.16 (0.20) |
| AGE vs. BCP | 0.545 (<0.0001) | 0.22 | −0.46 to 0.78 | 0.90 | 0.72–1.13 | −0.04 (0.31) |
| BCG vs. CZE | 0.826 (<0.0001) | −0.40 | −0.88 to −8.88 | 1.14 | 1.00–1.30 | 0.01 (0.18) |
| BCG vs. BCP | 0.791 (<0.0001) | −0.41 | −1.13 to 0.15 | 1.07 | 0.89–1.28 | −0.20 (0.24) |
| BCP vs. CZE | 0.624 (<0.0001) | 0.009 | −0.75 to 0.65 | 1.09 | 0.86–1.34 | 0.22 (0.28) |

Notes.
[a]Constant error is present.

**Table 2** Passing-Bablok regression analysis, Bland Altman analysis, and Spearman's/Pearson's correlation of electrophoresis fractions quantitated by AGE and CZE in serum from cattle; sample size: $n = 55$. Percentage of fraction values shown as median (interquartile range).

| Fraction | AGE | CZE | Spearman's/Pearson's correlation | Passing-Bablok regression | | | | Bland Altman bias |
|---|---|---|---|---|---|---|---|---|
| | Median (IQR) | Median (IQR) | $r_s$ value (p value) | y-intercept | 95% CI | Slope | 95% CI | Mean (SD) |
| Albumin (%) | 42.9 (4.28) | 40.7 (4.17) | 0.911 (<0.0001) | 0.22 | −0.02 to 0.43 | 0.97 | 0.90–1.06 | −2.4 (1.7) |
| Alpha 1 (%) | 7.0 (2.12) | 0.9 (0.3) | 0.207 (0.12) | 0.74 | 0.34–0.90[a] | 0.02 | 0.00–0.08[b] | −5.9 (1.3) |
| Alpha 2 (%) | 8.0 (1.27) | 14.5 (1.95) | 0.870 (<0.0001) | 2.60 | −1.40 to 5.25 | 1.50 | 1.16–2.00[b] | 6.7 (1.0) |
| Beta (%) | 10.5 (1.95) | 11.3 (1.57) | 0.864 (<0.0001) | 0.98 | −1.95 to 3.37 | 0.97 | 0.75–1.25 | 0.9 (1.1) |
| Gamma (%) | 32.2 (5.75) | 32.8 (5.07) | 0.977 (<0.0001) | 4.12 | 0.90 to 6.72[a] | 0.90 | 0.82–1.00 | 0.8 (1.3) |
| A/G ratio | 0.75 (0.13) | 0.69 (0.12) | 0.853 (<0.0001) | −0.03 | −0.15 to 0.06 | 1.15 | 1.00–1.33 | −0.07 (0.06) |

Notes.
[a]Constant error is present.
[b]Proportional error is present.

Electrophoretic separation of serum proteins revealed differences in fraction resolution between the methods. Using the AGE method, five fractions were quantified: albumin, alpha 1, alpha 2, beta, and gamma globulins. In contrast, the CZE method resolved six fractions: albumin, alpha 1, alpha 2, beta 1, beta 2, and gamma globulins, with beta globulins further subdivided into beta 1 and beta 2 (Fig. 3).

Comparison of protein fractions between AGE and CZE was also performed using Passing-Bablok and Bland-Altman analyses (Table 2). All fractions were significantly correlated with each other ($r = 0.85$–$0.91$, $p < 0.0001$), with the exception of the alpha 1 globulin fraction, which showed a weak and non-significant correlation ($r = 0.21$, $p = 0.12$). Constant error was observed for the alpha 1 and gamma globulin fractions, while proportional error was noted in the alpha 1 and alpha 2 fractions. Despite these discrepancies, strong agreement was found between the two methods for the majority of fractions, this is show in the Bland-Altman plots (Fig. 4), and the Passing-Bablok regression analysis (Fig. 5).

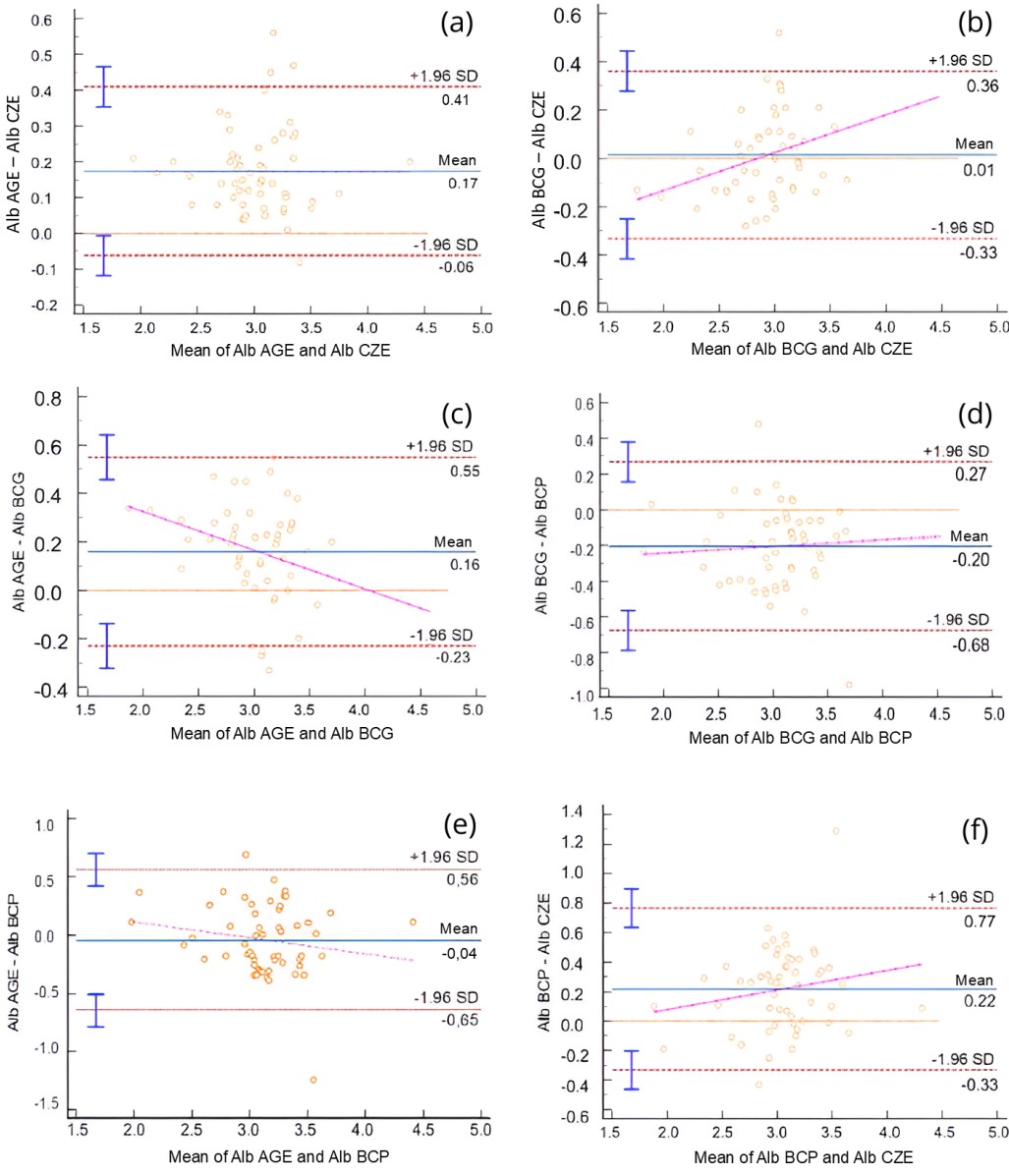

**Figure 1** **Bland-Altman plots comparing serum albumin (Alb) concentrations obtained using different analytical methods: agarose gel electrophoresis (AGE), capillary zone electrophoresis (CZE), bromocresol green (BCG), and bromocresol purple (BCP).** Each plot shows the difference between paired measurements (Y-axis) against their mean (X-axis). The solid blue line represents the mean bias; red dashed lines indicate the 95% limits of agreement (±1.96 SD). Vertical error bars represent standard error of the mean. A magenta regression line is shown when proportional bias was observed. (A) AGE *vs.* CZE, (B) BCG *vs.* CZE, (C) AGE *vs.* BCG, (D) BCG *vs.* BCP, (E) AGE *vs.* BCP, (F) BCP *vs.* CZE. Proportional bias was detected in panels (B), (C), (D), and (F), indicating method-dependent variability in albumin measurement.

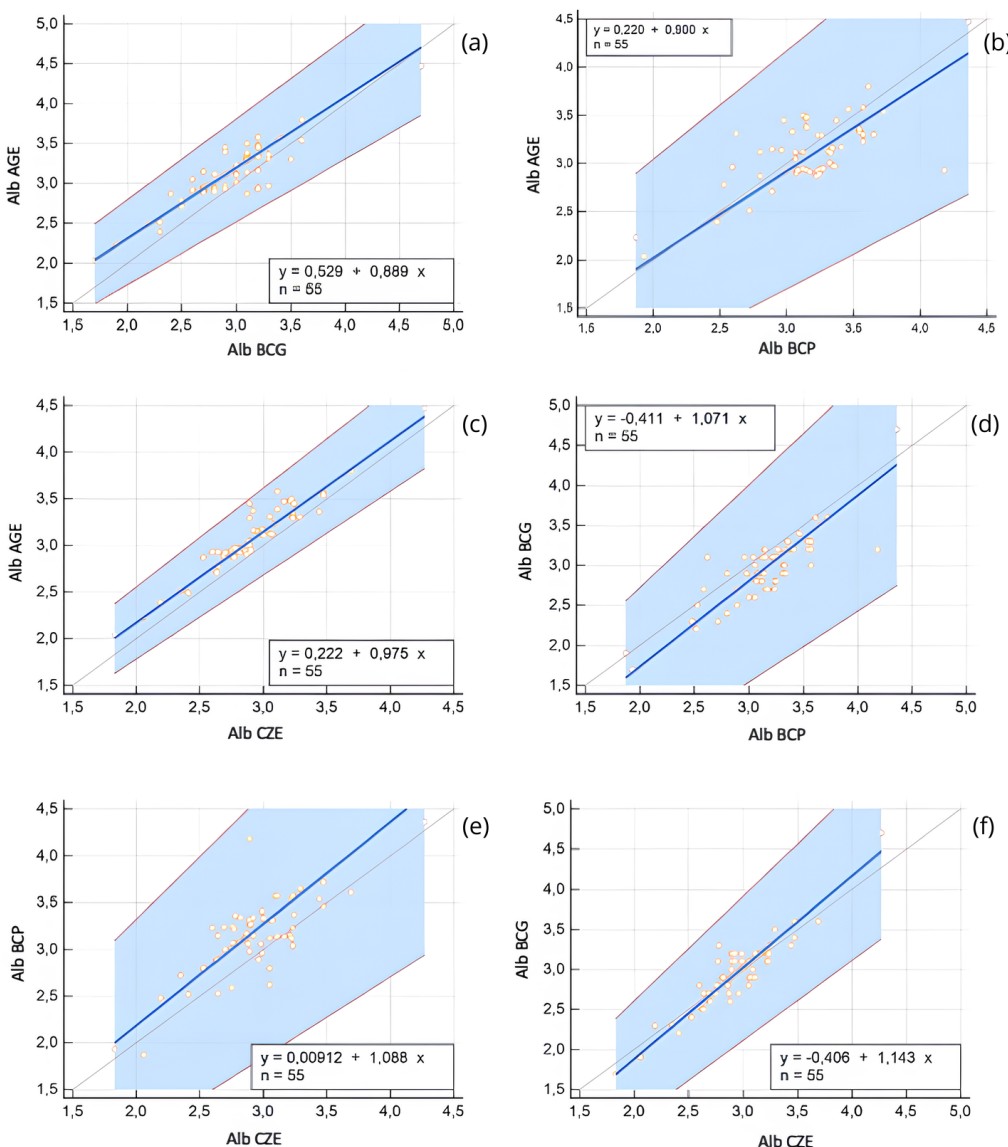

**Figure 2  Passing-Bablok regression plots comparing serum albumin (Alb) concentrations obtained using different analytical methods: agarose gel electrophoresis (AGE), bromocresol green (BCG), bromocresol purple (BCP), and capillary zone electrophoresis (CZE).** Each plot includes the regression line (blue), 95% confidence intervals (shaded blue area), and the line of identity (gray). The regression equation and sample size ($n = 55$) are shown within each panel. (A) AGE *vs.* BCG, (B) AGE *vs.* BCP, (C) AGE *vs.* CZE, (D) BCG *vs.* BCP, (E) CZE *vs.* BCP, (F) CZE *vs.* BCG. These comparisons reveal both constant and proportional differences between methods, emphasizing variability in albumin quantification depending on the assay used.

## Reference intervals

Reference intervals were established for all protein fractions obtained by AGE (Table 3) and CZE (Table 4) as well as for all albumin quantification methods (Table 5), in accordance with ASVCP guidelines. All data were transformed before analysis using the robust

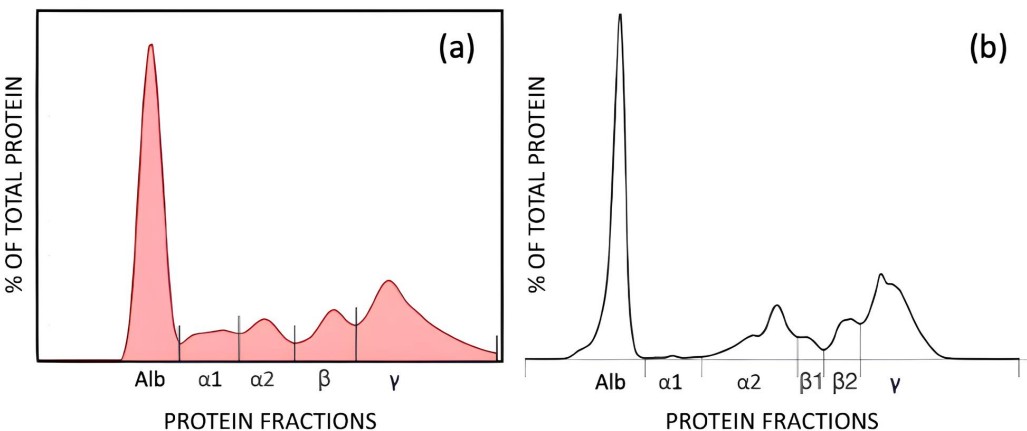

**Figure 3** **AGE (A), and CZE (B) electrophoretograms showing the fractions by each method.** AGE fractions correspond to albumin (Alb), alpha 1 (α1), alpha 2 (α2), beta (β), and gamma globulins (γ). CZE fractions correspond to albumin (Alb), alpha 1 (α1), alpha 2 (α2), β1 (β1), beta 2 (β2), gamma globulins (γ).

method. All outliers were retained as the population was defined as general health was well-established.

## DISCUSSION

Notably, all methods to determine albumin levels in cattle exhibited statistically significant correlations with each other. Remarkably, the bias across all methods was minimal. Albumin results by AGE showed a minor positive bias to BCG and a constant error was observed by Passing Bablok regression. This is contrary to previous reports of the overestimation of albumin by BCG proposed to be due to the reaction to globulins (*Keay & Doxey, 1984*; *Cray, 2021*; *Filípek & Illek, 2021*). This has been previously reported to be remediated by shortening the reaction time from 10 min to 30 s using manual and automated BCG methods (*Keay & Doxey, 1983*; *Filípek & Illek, 2021*). By comparison, BCG methods in the Ortho Vitros analyzer used in the present study have a 2.5-minute reaction time. The present study also did not corroborate previous studies regarding the utility of BCP showing just a minor positive bias to AGE results. While the specificity of the reaction between human albumin and BCP has been reported to be superior to BCG, its affinity for animal albumin including the cow has been reported to be considerably lower when using AGE as the standard (*Evans & Parsons, 1988*; *Filípek & Illek, 2021*). It is likely that these observed biases with BCG and BCP *versus* AGE are related to differences in the calibrators and reaction times in the chemistry analyzers as well as differences in the AGE methods.

Albumin was slightly overestimated by both BCG and BCP methods *versus* that quantitated by the CZE method. CZE, with the use of UV detection rather than dye binding, has been considered the gold standard for quantitation of albumin in human samples (*Christians et al., 2016*). In human serum samples, BCG method was also observed to overestimate albumin *versus* CZE (*Duly et al., 2003*). Only a minor positive bias was observed with the BCP method which is consistent with the bias observed in cow serum in

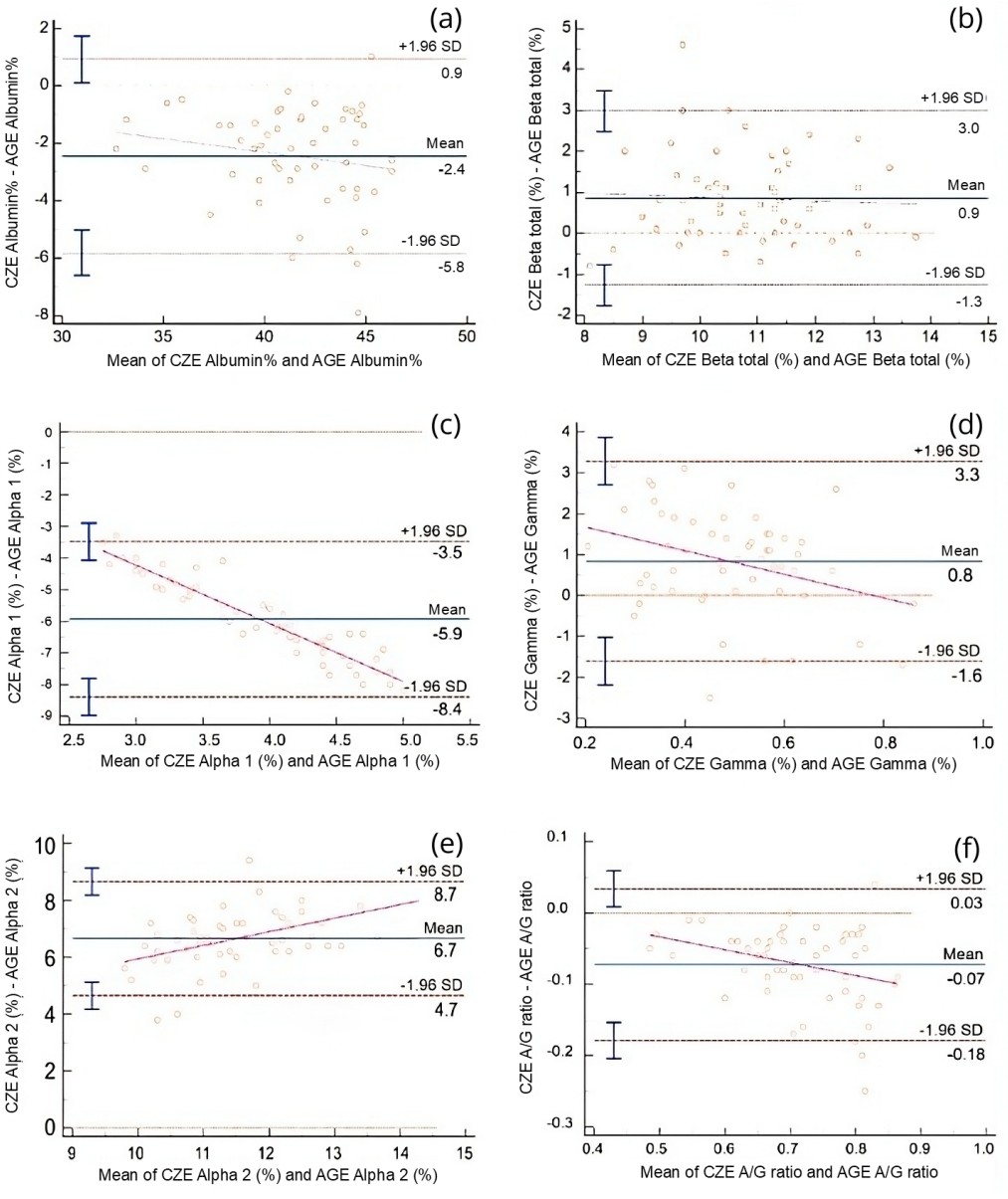

**Figure 4** **Bland-Altman plots comparing the relative (%) values of electrophoretic fractions obtained by capillary zone electrophoresis (CZE) and agarose gel electrophoresis (AGE), including the albumin-to-globulin (A/G) ratio.** Each plot displays the difference between CZE and AGE measurements (Y-axis) against their mean (X-axis). The solid line represents the mean difference (bias), while dashed lines indicate the 95% limits of agreement (±1.96 SD). Magenta regression lines indicate proportional bias when present. (A) Albumin (%), (B) Beta globulin total (%), (C) Alpha 1 globulin (%), (D) Gamma globulin (%), (E) Alpha 2 globulin (%), (F) A/G ratio. Marked biases and proportional differences are observed in alpha- and gamma-globulin fractions, indicating method-dependent variability that must be considered when interpreting protein electrophoresis results.

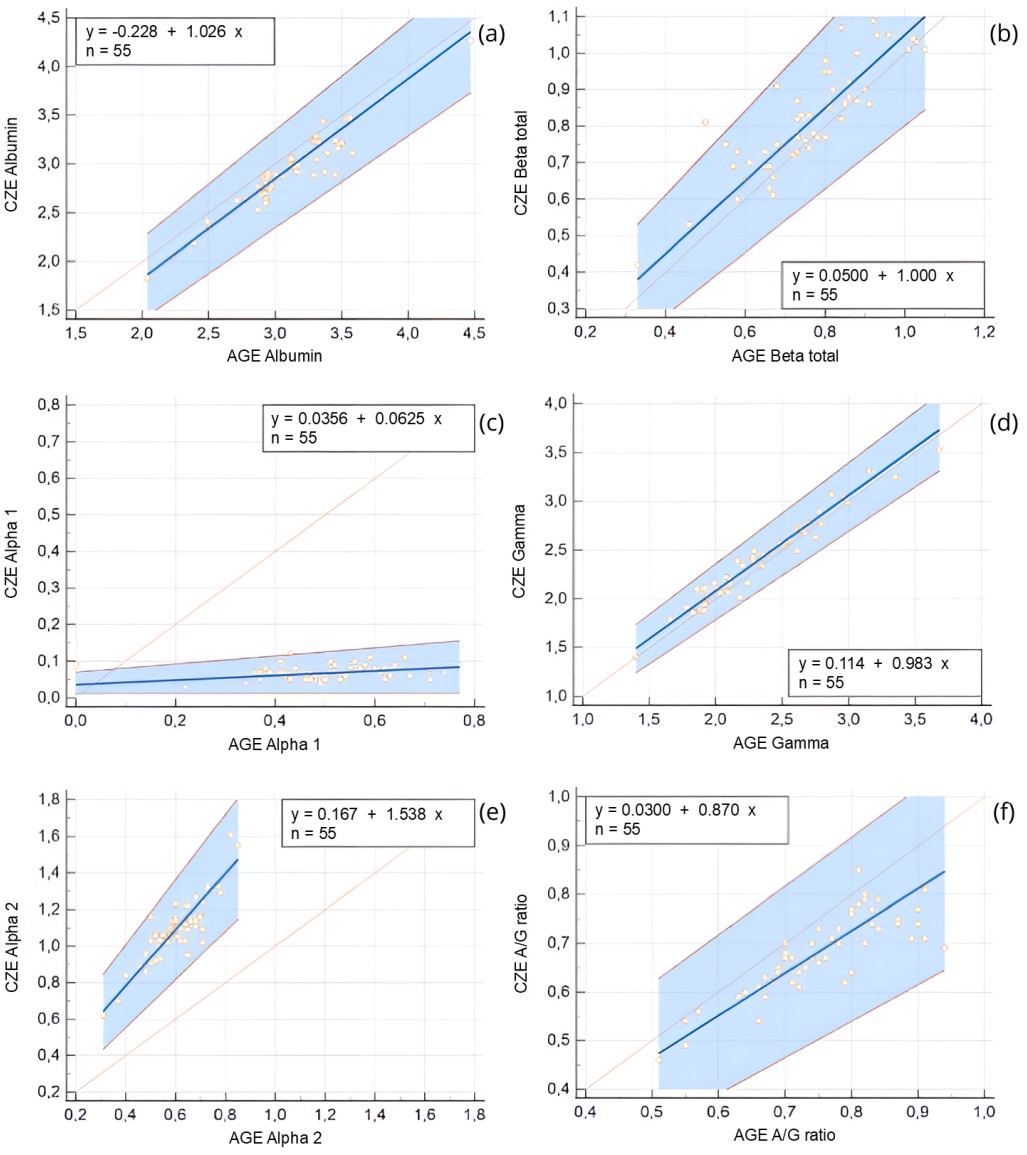

**Figure 5** **Passing-Bablok regression plots comparing capillary zone electrophoresis (CZE) and agarose gel electrophoresis (AGE) for protein fractions expressed as g/dL and for the albumin-to-globulin (A/G) ratio.** Regression lines (blue), 95% confidence intervals (blue shaded area), and identity lines (orange) are shown. Regression equations and sample size ($n = 55$) are reported in each panel. (A) Albumin (g/dL), (B) Beta globulin total (g/dL), (C) Alpha 1 globulin (g/dL), (D) Gamma globulin (g/dL), (E) Alpha 2 globulin (g/dL), (F) A/G ratio. The plots highlight differences in absolute protein quantification between methods, with larger deviations noted in alpha fractions and the A/G ratio.

the current study. The cause of this overestimation is difficult to identify but it may reflect the binding of globulins as proposed in the previous animal studies (*Keay & Doxey, 1984*; *Cray, 2021*; *Filípek & Illek, 2021*).

Overall, method comparison studies of albumin determination are challenging as there must be a designation of what is considered a gold standard assay. Although similar robust

studies have not been conducted with animal sera, CZE can be proposed to serve as the standard method given the level of automation and enhanced resolution over AGE (*Cray, 2021*). In support of this proposal, the AGE method showed a positive bias to CZE in the quantitation of albumin. This may reflect differences in these methods including protein fractionation methodology and the need for dye staining and densitometry in AGE (*Hooijberg et al., 2020*).

As dye binding methods are popular in automated clinical chemistry analyzers. results from these methods should be considered in the context of the clinical scenario. For example, in patients with ongoing acute phase responses there may be an excess of globulins such that BCG-based methods may produce false-positive results (*Garcia Moreira et al., 2018*; *Cray, 2021*). To this point, it is important to note that the samples examined in the present study were from clinically normal cattle; additional studies should be undertaken to better address method differences in cattle with ongoing inflammatory or infectious processes. In addition, there are also difficulties in the standardization and monitoring of albumin BCP assays for non-human samples because BCP, unlike BCG, does not strongly bind to bovine serum albumin, which is often present in calibrators and quality-control materials (*Garcia Moreira et al., 2018*).

Comparing the AGE and CZE methods, several differences were observed including the presence of constant error in alpha 1 and gamma globulin fractions as well as proportional error win alpha 1 and alpha 2 globulins. Overall, there was a very strong correlation between the methods except for the alpha 1 globulin fraction. Previous studies have proposed that certain proteins from the alpha fraction may migrate to the prealbumin fraction when employing capillary electrophoresis (*Flo et al., 2019*), however, no prealbumin fraction was detected in CZE method. *Alberghina et al. (2011)* and *Tóthová et al. (2013)* reported the absence of prealbumin identification using agarose gel electrophoresis. It can be argued that quantifying an ambiguously defined prealbumin migrating fraction introduces errors and biases in the quantification of other fractions. In general, the establishment of method-specific RI is crucial when employing either type of electrophoresis method (*Cray, 2021*).

Indeed, numerous researchers have observed significant variations in serum protein fractions across different domestic animals (*Alberghina et al., 2011*). The composition, configuration, and quantity of these fractions and subfractions vary greatly depending on the animal species and breed (*Fayos et al., 2005*). The most prominent discrepancies are found in the beta and gamma globulins. *Kaneko (2008)* reported only four fractions in bovine serum using cellulose acetate electrophoresis, which include albumin, alpha, beta, and gamma globulins. However, *Nagy et al. (2015)* employed agarose gel electrophoresis and identified six fractions in bovine serum, consisting of albumin, alpha 1 and alpha 2 globulins, beta 1 and beta 2 globulins, and gamma globulins. Furthermore, *Alberghina et al. (2011)*, *Piccione et al. (2012)*, and *Dede et al. (2014)* separated bovine serum proteins into five fractions, comprising albumin, alpha 1, and alpha 2 globulins, beta, and gamma globulins. These variations highlight the importance of considering the specific analytical techniques employed when characterizing serum protein fractions in bovine and other domestic animals.
**Table 3** Reference intervals for AGE fractions established from 55 cattle.

| Measurand | Units | n | Mean | SD | Median | Min | Max | P-value | Distribution | Method | LRL of RI | URL of RI | CI of LRL | CI of URL |
|---|---|---|---|---|---|---|---|---|---|---|---|---|---|---|
| Total protein | (g/dL) | 55 | 7.30 | 0.90 | 7.40 | 4.30 | 10.30 | 0.01 | NG | R, T | 5.81 | 8.78 | 5.46–6.16 | 8.43–9.13 |
| A/G ratio | | 55 | 0.75 | 0.10 | 0.75 | 0.51 | 0.94 | 0.41 | G | R, T | 0.58 | 0.92 | 0.54–0.62 | 0.88–0.96 |
| Albumin | (%) | 55 | 42.75 | 3.53 | 42.90 | 33.80 | 48.60 | 0.08 | G | R, T | 36.9 | 48.6 | 35.6–38.3 | 47.2–49.9 |
| Albumin | (g/dL) | 55 | 3.11 | 0.39 | 3.12 | 2.04 | 4.47 | 0.02 | NG | R, T | 2.41 | 3.58 | 2.32–2.62 | 3.59–3.89 |
| Alpha 1 | (%) | 55 | 6.9 | 1.24 | 7.0 | 4.50 | 9.10 | 0.02 | NG | R, T | 4.84 | 8.95 | 4.36–5.33 | 8.46–9.43 |
| Alpha 1 | (g/dL) | 55 | 0.49 | 0.13 | 0.50 | 0.00 | 0.77 | 0.001 | NG | R, T | 0.28 | 0.71 | 0.22–0.33 | 0.66–0.76 |
| Alpha 2 | (%) | 55 | 8.16 | 0.87 | 8.0 | 6.60 | 10.30 | 0.25 | G | R, T | 6.72 | 9.60 | 6.38–7.06 | 9.26–9.94 |
| Alpha 2 | (g/dL) | 55 | 0.59 | 0.10 | 0.60 | 0.31 | 0.85 | 0.58 | G | R, T | 0.42 | 0.77 | 0.38–0.46 | 0.73–0.81 |
| Beta total | (%) | 55 | 10.39 | 1.38 | 10.50 | 7.40 | 13.80 | 0.89 | G | R, T | 8.12 | 12.6 | 7.59–8.66 | 12.1–13.2 |
| Beta total | (g/dL) | 55 | 0.76 | 0.14 | 0.76 | 0.33 | 1.05 | 0.52 | G | R, T | 0.51 | 1.01 | 0.45–0.57 | 0.95–1.06 |
| Gamma | (%) | 55 | 31.8 | 3.83 | 32.20 | 24.50 | 41.80 | 0.38 | G | R, T | 25.5 | 38.1 | 24.0–26.9 | 36.6–39.6 |
| Gamma | (g/dL) | 55 | 2.32 | 0.44 | 2.28 | 1.40 | 3.68 | 0.06 | G | R, T | 1.60 | 3.05 | 1.42–1.77 | 2.88–3.22 |

**Notes.**

A/G ratio, albumin/globulin ratio; SD, standard deviation; G, Gaussian; NG, non-Gaussian; R, robust; T, transformed; RI, reference interval; CI, 90% confidence interval; LRL, lower reference limit; URL, upper reference limit.

**Table 4  Reference intervals for CZE fractions established from 55 cattle.**

| Measurand | Units | n | Mean | SD | Median | Min | Max | P-value | Distribution | Method | LRL of RI | URL of RI | CI of LRL | CI of URL |
|---|---|---|---|---|---|---|---|---|---|---|---|---|---|---|
| Total protein | (g/dL) | 55 | 7.30 | 0.90 | 7.40 | 4.30 | 10.30 | 0.01 | NG | R, T | 5.81 | 8.78 | 5.46–6.16 | 8.43–9.13 |
| A/G ratio | | 55 | 0.68 | 0.08 | 0.69 | 0.46 | 0.85 | 0.35 | G | R, T | 0.53 | 0.82 | 0.50-0.56 | 0.79–0.85 |
| Albumin | (%) | 55 | 40.30 | 3.23 | 40.70 | 31.60 | 45.80 | 0.07 | G | R, T | 34.9 | 45.6 | 33.7–36.2 | 44.4–46.9 |
| Albumin | (g/dL) | 55 | 2.93 | 0.39 | 2.92 | 1.83 | 4.27 | 0.02 | NG | R, T | 2.23 | 3.47 | 2.15–2.45 | 3.42–3.72 |
| Alpha 1 | (%) | 55 | 0.96 | 0.22 | 0.90 | 0.60 | 1.60 | 0.04 | NG | R, T | 0.59 | 1.32 | 0.50–0.67 | 1.24–1.41 |
| Alpha 1 | (g/dL) | 55 | 0.07 | 0.01 | 0.07 | 0.03 | 0.12 | 0.39 | G | R, T | 0.03 | 0.11 | 0.03–0.04 | 0.09–0.11 |
| Alpha 2 | (%) | 55 | 14.82 | 1.30 | 14.50 | 12.20 | 18.90 | 0.21 | G | R, T | 12.67 | 16.9 | 12.1–13.2 | 16.4–17.5 |
| Alpha 2 | (g/dL) | 55 | 1.08 | 0.16 | 1.09 | 0.62 | 1.61 | 0.02 | NG | R, T | 0.80 | 1.35 | 0.74–0.87 | 1.29–1.42 |
| Beta total | (%) | 55 | 11.26 | 1.32 | 11.30 | 7.70 | 14.10 | 0.70 | G | R, T | 9.07 | 13.4 | 8.56–9.59 | 12.9–13.9 |
| Beta total | (g/dL) | 55 | 0.82 | 0.14 | 0.82 | 0.42 | 1.09 | 0.91 | G | R, T | 0.58 | 1.06 | 0.52–0.63 | 1.01–1.12 |
| Gamma | (%) | 55 | 32.65 | 3.42 | 32.80 | 25.70 | 41.40 | 0.64 | G | R, T | 27.0 | 38.2 | 25.7–28.3 | 36.9–39.6 |
| Gamma | (g/dL) | 55 | 2.38 | 0.42 | 2.39 | 1.40 | 3.53 | 0.33 | G | R, T | 1.68 | 3.08 | 1.52–1.85 | 2.92–3.25 |

**Notes.**

A/G ratio, albumin/globulin ratio; SD, standard deviation; G, Gaussian; NG, non-Gaussian; R, robust; T, transformed; RI, reference interval; CI, 90% confidence interval; LRL, lower reference limit; URL, upper reference limit.

Fonseca et al. (2025), *PeerJ*, DOI 10.7717/peerj.19685

**Table 5** Reference intervals for albumin BCG, and albumin BCP established from 55 cattle.

| Measurand | Units | *n* | Mean | SD | Median | Min | Max | *P*-value | Distribution | Method | LRL of RI | URL of RI | CI of LRL | CI of URL |
|---|---|---|---|---|---|---|---|---|---|---|---|---|---|---|
| Albumin BCG | (g/dL) | 55 | 2.95 | 0.45 | 3.00 | 1.70 | 4.70 | 0.0024 | NG | R, T | 2.21 | 3.69 | 2.03–2.38 | 3.52–3.87 |
| Albumin BCP | (g/dL) | 55 | 3.15 | 0.43 | 3.15 | 1.87 | 4.36 | 0.030 | NG | R, T | 2.44 | 3.87 | 2.27–2.60 | 3.70–4.04 |
| Albumin AGE | (g/dL) | 55 | 3.11 | 0.39 | 3.12 | 2.04 | 4.47 | 0.02 | NG | R, T | 2.41 | 3.58 | 2.32–2.62 | 3.59–3.89 |
| Albumin CZE | (g/dL) | 55 | 2.93 | 0.39 | 2.92 | 1.83 | 4.27 | 0.02 | NG | R, T | 2.23 | 3.47 | 2.15–2.45 | 3.42–3.72 |

**Notes.**

BCG, bromocresol green; BCP, bromocresol purple; AGE, agarose gel electrophoresis; CZE, capillary zone electrophoresis; SD, standard deviation; G, Gaussian; NG, non-Gaussian; R, robust; T, transformed; RI, reference interval; CI, 90% confidence interval; LRL, lower reference limit; URL, upper reference limit.

The alpha fraction is known to exhibit the fastest migration among all the globulins, typically appearing as alpha 1 and alpha 2 subfractions in most species (*Alberghina et al., 2011*). In our study, both AGE and CZE methods showed the migration of alpha globulins into these two zones, but the concentrations of these fractions differed significantly for alpha 1 globulin where the CZE showed a very low protein composition. In a previous AGE report of Modicana cows, these two fractions constituted 3.59−7.13% for alpha 1 globulin and 5.09–13.96% for alpha 2 globulin of the total serum proteins which is similar to that calculated in the current study (*Alberghina et al., 2011*). This finding is consistent with other previous studies using AGE (*Dede et al., 2014*; *Nagy et al., 2015*), but not for the present CZE results. The conflicting data can be attributed to the use of different electrophoretic techniques for separating protein fractions and the sensitivity and increased resolution of the CZE method (*Luraschi, Dea & Franzini, 2003*). Notably, when analyzing the current data as one total alpha fraction (*i.e.,* a sum of alpha 1 and 2 fractions), AGE and CZE exhibit a strong correlation.

For all the other fractions, AGE and CZE yielded correlated results, while AGE showed slight differences compared to other studies. As with other mammals, albumin is the most prominent serum protein in the electrophoretogram of cattle. Previously, using AGE methods, albumin ranged from 39–58% of total protein (*Alberghina et al., 2011*). In the present study, the values of albumin were slightly lower than this former report but similar to *Nagy et al. (2015)*, and higher than those reported by *Dede et al. (2014)* for healthy cattle. Using AGE methods, the beta globulins were reported to represent 7.38–15.28% of the total protein (*Alberghina et al., 2011*). In the present study, the values of beta fraction were similar to this report as well as that shown by *Dede et al. (2014)* and slightly lower than those reported by *Nagy et al. (2015)*. The gamma globulin fraction constituted $31.8 \pm 3.8\%$ (mean $\pm$ SD) of the total serum protein concentration for AGE, which was similar to the findings of *Dede et al. (2014)* but higher than those reported by *Alberghina et al. (2011)* and *Nagy et al. (2015)*. The A/G ratio was slightly lower than that reported by *Alberghina et al. (2011)* and *Nagy et al. (2015)* but higher than that reported by *Dede et al. (2014)*. These differences may be related both to the electrophoresis method used and how it was implemented as well as the breed and age of the cattle.

The composition and migration characteristics of globulin fractions exhibit significant diversity. In general, CZE has demonstrated advantages in terms of ease of use, higher resolution, and lower coefficient of variation compared to AGE methods (*Bossuyt, 2006*). CZE enables enhanced detection levels, allowing for the resolution of certain fractions, and these findings align with previous reports in various mammalian species (*Crivellente, Bonato & Cristofori, 2008*; *Giordano & Paltrinieri, 2010*). Studies investigating CZE applications in humans, as well as limited studies in other mammalian species, have reported finer separation of alpha and beta globulin fractions (*Roudiere et al., 2006*; *Crivellente, Bonato & Cristofori, 2008*). In the present study, there was a consistent separation of beta 1 and beta 2 globulins in CZE which was not possible by the AGE method.

Serum protein electrophoresis is a valuable tool for evaluating various species (*Cray, 2021*). The preliminary RI presented in this study were derived from a limited number of healthy cattle. Although the results seem comparable to those reported for other ruminant

species, further investigations are necessary to determine if there are breed-specific differences or influence of age and sex in the RI for serum total protein concentration and subsequent protein fractions. Conducting additional studies will contribute to the development of more robust intervals and the examination of electrophoretic abnormalities in clinically abnormal bovines, thereby enhancing the understanding of clinical applicability of these tools.

## CONCLUSION

Dye-binding assays, such as BCG and BCP, deliver reliable results and offer an alternative due to their ease of automation, speed, and cost-effectiveness. Nevertheless, electrophoresis is the method of choice for the analytical separation of proteins, including the determination of albumin levels. It should be noted that while CZE offers improved resolution, performing these analyses presents specific challenges in determining fraction boundaries, establishing RI, and interpreting the results. Overall, consideration should always be given to the method and analyzer specific RI when working with animal samples.

### Funding

This work was funded by the Coordination of Superior Level Staff Improvement, CAPES, Brazil. The funders had no role in study design, data collection and analysis, decision to publish, or preparation of the manuscript.

### Grant Disclosures

The following grant information was disclosed by the authors:
Coordination of Superior Level Staff Improvement, CAPES, Brazil.

### Competing Interests

The authors declare there are no competing interests.

### Author Contributions

- Leandro Abreu da Fonseca conceived and designed the experiments, performed the experiments, analyzed the data, prepared figures and/or tables, authored or reviewed drafts of the article, and approved the final draft.
- Fabiano Montiani-Ferreira conceived and designed the experiments, performed the experiments, analyzed the data, prepared figures and/or tables, authored or reviewed drafts of the article, and approved the final draft.
- Marilyn Rodriguez performed the experiments, prepared figures and/or tables, and approved the final draft.
- Joao Henrique Jabur Bittar performed the experiments, prepared figures and/or tables, and approved the final draft.
- Fabricia Modolo Girardi performed the experiments, authored or reviewed drafts of the article, and approved the final draft.

- Theo Matos Arantes Moraes performed the experiments, authored or reviewed drafts of the article, and approved the final draft.
- Carolyn Cray conceived and designed the experiments, performed the experiments, analyzed the data, prepared figures and/or tables, authored or reviewed drafts of the article, and approved the final draft.

## Animal Ethics

The following information was supplied relating to ethical approvals (i.e., approving body and any reference numbers):

Ethics committee on Animal Use of Federal University of Vicosa approval 56/2024.

## Data Availability

The raw data is available in the Supplementary File.

## Supplemental Information

Supplemental information for this article can be found online at http://dx.doi.org/10.7717/peerj.19685#supplemental-information.

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
