# Peer review of "Measurement of bovine (Bos taurus) serum albumin by different methods and the comparison of protein fractions determined by capillary zone electrophoresis and agarose gel electrophoresis"

_PeerJ, doi:10.7717/peerj.19685_

## Round 0.1 · original submission · Major Revisions

Please consider the reviewers' suggestions and submit a revised version along with a point-by-point response letter that addresses all concerns.

Reviewer 1 ·

Basic reporting

The manuscript deals with an interesting and important area of laboratory diagnostics, since the protein profile within blood biochemistry is a frequent part of blood analyses in animals. It certainly brings significant knowledge that can be used in clinical practice. However, from my point of view, it would be appropriate to make some additions and adjustments so that the achieved results are better presented and clearer, and explained. Therefore, I suggest some adjustments or changes:

- When citing multiple papers in the text, it would be appropriate to arrange them by year from oldest to newest

Experimental design

- Since the work also involves determining reference intervals that could be influenced by various factors, it would be appropriate to supplement the data on dairy cows with the period or stage of reproduction, whether they were only non-pregnant, or whether it was a set of animals also in different periods of pregnancy

Validity of the findings

- The manuscript lacks a Conclusion, both at the end of the paper and in the abstract, which would briefly summarize the authors' view of the performed observations and achieved results

Additional comments

- The manuscript presents 5 tables and 5 graphs, but their description within the results is, in my opinion, very brief and insufficient. There is no commentary, there are basically only references to tables and graphs, as if the authors left it up to the reader to evaluate the achieved results themselves. A lot of data is presented, a lot of statistical analyses, but without their description by the authors of the work within the results section of the work. What is stated within the brief description of the results is half a repetition of the facts stated in the Statistical analysis (Materials and Methods) section. The results section describes only Figure 3 in 1 sentence – the difference in protein fractions between AGE and CZE. Therefore, it is requested to omit the repetition of methodological issues and describe the results, and the authors' opinion on the achieved results.

·

Basic reporting

The present manuscript compares methods for determining the albumin concentration in cattle. Moreover, the authors described the importance and necessity of this context and the importance of the statistical evaluation. In this sense, I consider the publication with minor corrections. The abstract is recommended to incorporate a direct implication of this research in the livestock industry.

Experimental design

-

Validity of the findings

The sample size is adequate. However, it is necessary to mention that the implications of the results for using organisms at other ages or the influence of sex.

Additional comments

Abstract:
Define ASVCP guidelines
It is recommended to incorporate a conclusion.
Introduction
Line 39.- The bromocresol green (BCG) and bromocresol purple (BCP) are double defined.
Line 48.- “,”

Methods
Lines 92 and 102.- The sentences “Plasma samples were diluted 1:2 using PBS 1X buffer as
diluent” and “samples were diluted 1:8 using urine running buffer as diluent” are redundant.
Line 117.- Negligible < 0.29?

Results
Figure 1.- In the first panel, the purple line is not observed.
Figure 2.- The title axis has a different letter size.
Figure 3.- It is recommended that the denomination of each fraction be on the figure.

---

## Round 0.2 · Minor Revisions

Please restore Figure 1 to its publication-quality resolution and clarity for identifying the line colors.

Reviewer 1 ·

Basic reporting

No comment to this area.

Experimental design

No comment to this area.

Validity of the findings

No comment to this area.

Additional comments

Based on the assessment of the modifications and changes made in accordance with the proposals, I can conclude that the proposed modifications have been made and therefore I have no further comments on the submitted manuscript.

·

Basic reporting

The industrial implications of the present research are more evidence. I consider that it has the quality for publication. However, only a detail was identified, and it is mentioned in follow section.

Figure 1, after its modification, some points and purple lines were faded. It promotes the reduction of the publication quality.

Experimental design

-

Validity of the findings

-

Additional comments

-

---

## Round 0.3 · accepted · Accept

Thanks for addressing all the reviewer suggestions.